# Positional Features of the Mandibular Condyle in Patients with Facial Asymmetry

**DOI:** 10.3390/diagnostics13061034

**Published:** 2023-03-08

**Authors:** Diego Fernando López, Valentina Rios Borrás, Rodrigo Cárdenas-Perilla

**Affiliations:** 1Orthodontics Department, Universidad del Valle, Cali 760043, Colombia; 2Escuela de Odontología, Universidad del Valle, Cali 760043, Colombia; 3Nuclear Medicine Department, Clinica Imbanaco-Grupo Quirón Salud, Cali 760043, Colombia

**Keywords:** computed tomography, cone beam computed tomography, condylar anatomy, facial asymmetry, temporomandibular joint

## Abstract

Objective: To describe the position of the mandibular condyle, the size of the joint spaces and the condylar angulation in patients with facial asymmetry (FA), and to classify these results according to the type of FA and compare them with a reference group without FA. Materials and Methods/Patients: An observational, cross-sectional, descriptive study using computed tomography (CT) was conducted on a sample of 133 patients with a clinical diagnosis of FA derived from the following entities: hemimandibular elongation (HE) (*n* = 61), hemimandibular hyperplasia (HH) (*n* = 11), condylar hyperplasia in its hybrid form (HF) (*n* = 19), asymmetric mandibular prognathism (AMP) (*n* = 25), glenoid fossa asymmetry (GFA) (*n* = 9) and functional laterognathism (FL) (*n* = 8). Likewise, a group of 20 patients without clinical or tomographic characteristics of FA was taken and their complete cone beam tomography (CBCT) scans were analyzed. The quantified variables were joint spaces (anterior, middle and posterior), angle of the condylar axis and condylar position. All measurements were performed using the free, open-source Horos software. Results: Most of the subjects without FA had a right middle condylar position (55%), while in the patients with FA the anterior condylar position predominated. On the left side, the most frequent condylar position was anterior, including the group without FA, except in the HH group. Considering the measurements of the anterior, middle and posterior joint space (mm) on the right side (anterior JS: 1.9 mm, middle JS: 2 mm and posterior JS: 2.8 mm) and on the left side (anterior JS: 2.7 mm, middle JS: 2.1 mm and posterior JS: 2.6 mm) of the subjects without FA, compared to those with FA, the latter presented smaller distances in all diagnoses and only for the right posterior JS (1.9 mm) in HH, was not significant. The condylar axis of the AF group showed significant differences with smaller angles for the left side in those diagnosed with HE (65.4°) and HH (56.5°) compared to those without AF (70.4°). Conclusions: The condylar position of patients with FA tends to be anterior, both on the right and left sides, while for cases without FA it is middle and anterior, respectively. Patients with FA have smaller joint spaces (mm) compared to patients without FA, with the exception of HH for the right posterior JS.

## 1. Introduction

The temporomandibular joint (TMJ) is a bilateral, synovial, ginglymoarthrodial structure. It has in common with other articulations of this kind the anatomic conformation with two articular surfaces (the glenoid fossa of the temporal bone and the mandibular condyle), an articular disc, the articular capsule, ligaments and synovial liquid. A distinctive characteristic of this joint is that its articular surfaces are covered by fibrocartilage [1,2,3]. The TMJ is a highly functional demand articulation and therefore it is susceptible to present painful symptoms of different etiologies, including muscular and capsular alterations, ligament disorders, altered masticatory patterns and changes in the position of bone components and the articular disk. Therefore, under altered anatomic conditions, functional changes causing temporomandibular disorders (TMD) and/or inner articular damage are expected [3].

Although the TMJ has a high adaptive ability, the anatomic position may be affected by pathologic entities that generate substantial morphologic changes in facial expression, as is the case of facial asymmetry (FA) related to unilateral condylar hyperplasia (UCH) [4].

UCH is due to excessive growth of a mandibular condyle generated by bone metabolic hyperactivity. It is a self-limited condition, frequently unilateral, with esthetic, occlusal and functional consequences derived from the change in mandibular position [4,5].

Another entity altering the position of the articular fossa is glenoid fossa asymmetry (GFA), which is evident during the first years of development and is a defect in the proliferation, migration and differentiation of neural crest cells [6,7].

Adaptive remodeling following a severe trauma is another possible cause of asymmetry with no alteration of the mandibular condyle [8].

The mandibular anatomy may be altered as well by mandibular asymmetric prognathism (MAP), due in this case to a bilateral difference in the effective size of the mandible. MAP etiology is genetic, and it is evident during the first stages of dental development and develops skeletal Class III [9].

Finally, functional laterognathism (FL) is an entity causing FA related to changes in mandibular position appearing early at the occlusal level and characterized as a secondary adaptation of the mandible to a disbalance in the skeletal and occluso-maxillo-mandibular relationship. This secondary adaptation, if it is not early treated negatively evolves during growth to a true skeletal asymmetry with no differences in size of the condyle skeletal components or in the mandibular ramus [10,11].

Although some authors [12,13,14] suggest that the inner TMJ deterioration and a severe TMD may be predisposing factors to asymmetry in mandibular position, the relationship between the pathologies generating FA and the presence or absence of TMD is not well established because some alterations are present with no evidence of articular signs or symptoms [15].

The literature reports TMD patients without FA, presenting changes in the position of the mandibular condyles, characterized by a more posterior displacement of them [16]. Regarding dimensional changes in the articular spaces, some authors associate the reduction in superior and posterior space, as well as the increment in the anterior space, to anterior displacement of the disk in patients with no significant FA [17,18]. However, condylar position and the size of articular spaces in relation to TMD is a controversial subject. In patients with asymmetry, no reports were found indicating a significant difference in the angle formed by the latero-medial plane of each condyle and the mid-sagittal plane (MSP) [19].

The most effective way to evaluate the position of all the TMJ components is through a tomographic image able to detect sagittal, coronal and axial changes [20]. Therefore, the objective of this study was to use computed tomography (CT) in a group of patients with FA and cone beam computed tomography (CBCT) in non-asymmetric subjects, to obtain linear and angular measurements of the mandibular condyle position with respect to the articular cavity.

## 2. Materials and Methods

There was no risk research, using only retrospective documental data with no intervention. This study was approved by the Institutional Ethics Committees involved (Clínica Imbanaco: CEI-545 and Universidad del Valle: 032-021) and it was conducted according to the principles of the Declaration of Helsinki.

The CT data (Figure 1) of 133 patients treated in a clinical center of high complexity (Imbanaco) during January 2015 and January 2020 were evaluated. The inclusion criteria were diagnosis of FA and complete and acceptable CT images. The exclusion criteria were antecedents of TMJ pathology and/or surgery, trauma or fracture, treatment with occlusal splints, orthognathic surgery, dentofacial syndromic anomalies, arthritis and incomplete CT studies.

For UCH cases, the affected side was defined as the side with condylar overdevelopment; for MAP and FL, it was the side of mandibular deviation and for GFA it was the side with evident upper projection of the articular cavity.

CT images were obtained with PET/CT Biograph mCT20 (Siemens, Erlangen, Germany) equipment. Cranial images were obtained without contrast media, from vertex to sternal fork, applying the following parameters: section thickness 0.75 mm, pitch 1.0 and cubic matrix 512 × 512, isotropic voxel (size: 0.58 × 0.58 × 0.87 mm) to avoid image distortion in adult and growing patients. CT images were reconstructed using a B26F homogeneous, low-dose filter for anatomic location. All the patients were positioned with fixed head to avoid movement artifacts and facilitate image fusion.

The CBCT images (Figure 2) of 20 patients scheduled to initiate orthodontic treatment, with no mandibular deviation, suspected FA or TMD signs, obtained from April 2019 to March 2022, were selected from the Oral Radiology Department of the Universidad del Valle, Cali, Colombia.

CBCT images were obtained with i-CAT 17-19 equipment. Cranial images were obtained with no use of contrast media, from nasion to menton. The patients were in corrected natural head position. The following parameters were applied: camp window (FOV): 16 cm, width 0.250 mm, isotropic voxel (size: 0.25 × 0.25 × 0.25 mm) to avoid any image distortion in adult and growing patients.

The CT and CBCT images were stored in digital form and digital communications in medicine (DICOM). The DICOM 2D images were downloaded to the Horos software for processing, visualization and bidimensional measurement of the anatomic structures as described in Table 1.

The measurements were registered by an operator expert in the software management and TMJ anatomy. Each set of images and data was evaluated and classified under operator and clinician agreement, according to the craniofacial characteristics of the asymmetry [23], (Table 2).

### Statistical Analysis

Descriptive data were presented as central tendency (mean, median) and standard deviation or P25-P75, following the Shapiro–Wilk normality test for parametric variables. Non-parametric variables are expressed as absolute and relative percentage frequency.

Initially the intraoperator agreement was estimated by the correlation coefficient of agreement (CCA), obtaining a CCA value of 89% for the right anterior space and >90% for the other data and the condylar axis angle. Comparative tests (chi-square, *t*-test or U test) were applied as necessary. Any *p* value < 0.05 was accepted as significant. The statistical program used was R 4.2.2.

## 3. Results

Considering the selection criteria, a database of 133 patients with a diagnosis of FA was obtained. Additionally, the data of 20 non-asymmetric orthodontic patients with no FA or signs/symptoms of TMD were included. The median age for the non-asymmetric group of subjects was 22 years, and for the FA group age was in a range of 14–26, with the lower median (14 years) in the GFA group and the highest median (26 years) in the HH group. However, 75% of the patients were under 30 years. Female gender represented 61.4% of the total sample. The right side was more frequently affected (51.4%) and the more frequent diagnosis was HE, representing 45.9% of the asymmetry group. (Table 3).

According to the classification published by López et al. [23], the kind of FA was established as: condylar hyperplasia (CH): 91 cases (61 HE, 11 HH and 19 HF), MAP: 29, GFA: 9 and FL: 8.

In Table 4, the sample is regrouped according to the condylar position: posterior: <12%, middle: −12 to 12% and anterior >12%.

It was found that for the right-side data most non-asymmetric subjects had middle position (55%), while in patients with FA diagnosis the anterior condylar position was the most frequent, between 44 to 91% depending on the kind of FA. On the left side for both groups (FA and no FA), the anterior condylar position was the most frequent, except in the HH group.

Regarding the condylar position (%) in both sides, when the FA patients were compared to the without FA group (right: 7.4% and left: 11.7%) significant differences were found in the MAP group (15.7%). However, when comparing only the affected side, the differences were not significant in MAP patients. In the HH group, compared to the without FA group, the difference is significant for the right side (33.3%, *p* < 0.05) and in general (*p* < 0.01). In the GFA group, there was significant difference in the left side (30.8%, *p* < 0.05). (Table 5).

Table 6 shows that, comparing the measurements of the condylar axis (°) of non-asymmetric subjects (right: 68.9° and left: 70.4°) versus patients with FA, there are only significant differences in measurements on the left side of those diagnosed with EH (65.4°) and HH (56.5°), without disaggregating by affected side.

Taking into account the measurements of the anterior, middle and posterior joint space (mm) on the right side, it is observed that when comparing the non-asymmetric subjects (anterior JS: 1.9 mm, middle JS: 2 mm and posterior JS: 2.8 mm) with the measurements of patients with FA, the latter present smaller distances with statistically significant differences in all diagnoses and only for the posterior joint space (1.9 mm) in HH, it is not significant. Additionally, when they are analyzed by the affected right side, statistically significant differences are found in most entities, with the exception of the GFA, which has a sample size of only 4 cases.

For the measurements of the left side, it is observed that when comparing the non-asymmetric subjects (anterior JS: 2.7 mm, middle JS: 2.1 mm and posterior JS: 2.6 mm), with those diagnosed with asymmetry, there are also significant differences in all entities, presenting the latter shorter distances. When disaggregating with respect to the left affected side, all those diagnosed with FA have minor joint spaces with statistically significant differences and only the posterior joint space of the GFA (2.1 mm), FL (2.1 mm) and HH (2 mm), were not significant, however, there are samples of less than 5 in these cases (Table 7).

## 4. Discussion

The spatial orientation of the mandibular condyle with respect to the joint cavity in the TMJ may be influenced by anatomical, functional and/or pathological characteristics [25]. Some studies have evaluated these characteristics and their relationship with joint disorders or TMD, but few have focused on patients with structural skeletal disorders such as patients with FA. In the present study, the characteristics related to the condyle and its articular cavity in patients with different entities causing FA were evaluated.

With respect to joint spaces, Ikeda et al. [24] determined in their study with CBCT mean values in non-asymmetric patients, where the anterior, middle and posterior JS were 1,3, 2,5 and 2,1 mm, respectively. These values are close to those found in non-asymmetric patients in the present study, and far from those found in asymmetric patients. In this study, it was evidenced that the population with FA, regardless of the entity that produces the alteration, has smaller joint spaces than those patients without FA. Regarding this, Major et al. [25] reported that alterations in the joint spaces were associated with anterior displacement of the disc and a decrease in its length, although in their study of growing patients, the decrease in joint space was limited to the medial space. Likewise, A. K. Bag et al. [26] reported the possible association between the decrease in joint spaces with unilateral and bilateral disc displacements.

This means that if the function of the articular disc, in addition to supporting joint loads, is to provide synovial fluid to the bone surfaces that helps its nutrition, oxygenation, lubrication and hydration [27], the possible displacement and alteration in its anatomy would mean equally pathological and functional changes [28].

In fact, it has been hypothesized that the reduction in the joint space affects the condylar position in the contralateral TMJ [29]. However, the assessment of the joint space by itself is not enough to determine whether or not there is presence of TMD [30].

Likewise, in the present study, when the values are analyzed by affected side and by each entity, with the exception of GFA, all joint spaces were smaller. It was even evident in cases of condylar hyperplasia, in which there is a substantial change in the condylar size and the height of the joint cavity towards the affected side [31]. It was evidenced that the joint spaces were decreased with respect to the non-asymmetric subjects and only for the posterior joint space it was not significant in HH. The lack of significance may be explained by the small number of cases (n = 4).

Regarding the demographic characteristics, the majority of patients with FA were women and the most affected side was the right side. This is coincident with prior studies published by Raijmakers et al. [32] and López et al. [33].

The condylar position showed a higher percentage of middle condylar position in the right condyle for non-asymmetric subjects, while for asymmetric patients it was predominantly anterior, independent of the kind of FA. An interesting observation was that the right side was affected in 54.1% of the patients. On the other hand, when the left side was analyzed, all presented an anterior condylar position, including the non-asymmetric subjects, and it was not evident only for the four cases of HH. Similar results with differences between sides were obtained by Chae et al. [34] in an adolescent population and with a predominance of anterior condylar position in the left joint in the study of Ganugapanta et al. [35].

The comparison of each entity of FA and the without FA subjects was significant in MAP and HH groups only for the right side. The position in these cases was anterior as well. The lack of coincidence between sides and between with or without FA groups is coincident with the reports published by Paknahad et al. [36] and Guerrero et al. [37], showing that there are no differences in the condylar position in patients with or without TMD. Additionally, Lelis et al. [38] did not find differences between symptomatic and asymptomatic patients; as reported by Choi et al. [39], even in patients who underwent orthognathic surgery from sagittal mandibular osteotomies to correct mandibular prognathism and facial asymmetry, no changes in the condylar position were observed after surgery.

Differences have been reported for specific malocclusions such as the anterior open bite and posterior cross-bite, which show posterior condylar positions. [40]. Skeletal discrepancies in Class II subjects, show antero-superior condylar positions and hyperdivergent patterns with higher risk of condylar displacement [41,42].

In patients with asymmetry and a resulting posterior crossbite, as is the case with HE and FH, and which can also occur in cases of PMA, LF and even in GFA, Almaqrami et al. [43] postulate that skeletal crossbite is accompanied by morphologic and positional features in the TMJ associated with dental unilateral posterior crossbite and are associated with specific asymmetry on one side of the TMJ. In the present study differences between sides were not significant, but the condylar position was measured only in the sagittal plane, not in the transaxial.

In relation to the condylar axis, Westesson et al. [44] described a more closed axial condylar angle in normal TMJs, while for affected joints, such as those with disc displacement, this angle was much more open. Regarding this, Al Rawi et al. [30], found that there were differences between men and women for the angle of the condylar axis, being more closed in women. Unlike the findings of Westesson et al. [44], however, the angle of the condyle axis tended to decrease significantly in patients with TMD, both for men and women, showing internal rotation of the condyle in affected TMJs.

In the present study, differences were only found with respect to the non-asymmetric subjects for HE and HH in the left condylar axis in general, presenting smaller angles, but when disaggregated by the affected side, no differences were found. In this regard, it is worth mentioning that the universe of the present sample was patients with FA and not TMD. Similar results are reported by Rodrigues et al. [19] evaluating the angle between the latero-medial plane of each condyle and the mid-sagittal plane in Class I patients with no FA. The bilateral comparison of this angle shows mean values very similar to those of the non-asymmetric subjects in the present study (right side 70.10° and left side 69.96°).

Although it is well accepted that CT and CBCT imaging are gold standards for assessing morphologic and structural features of craniofacial bones and TMJ [45,46], they lack sensitivity for assessing soft tissues that are relevant for describing TMD. [47]. Therefore, it is suggested that future studies include joint symptoms and correlate AF with TMD. One limitation of this research is that the slice thickness of medical tomographies (0.75 mm) is an unmodifiable characteristic of the medical center since they are standardized both for accuracy and for radiation dose to this measure and cannot be with smaller slice thicknesses that, although they give more image sharpness, increase radiation.

## 5. Conclusions

There are marked differences between the sides in condylar position, both in patients with FA and without FA. The right side tends to have a middle position in non-asymmetric subjects and an anterior position in all FA patients. The left side has a predominantly anterior position in both AF and non-AF cases.

A greater anterior condylar position was evidenced for the right side in MAP (*p* = 0.04); for the affected left side in GFA (*p* = 0.03); and both general and for the affected right side in HH (*p* < 0.01 and *p* = 0.03), respectively, compared to the group without AF.

Patients with FA have reduced anterior, middle and posterior joint spaces with respect to non-asymmetric patients for both the right and left joints. There were no significant differences only in the right posterior joint space of the HH.

The angle of the condylar axis only showed differences for the HE and HH on the left side, these being smaller with respect to the non-asymmetric ones.

## Figures and Tables

**Figure 1 diagnostics-13-01034-f001:**
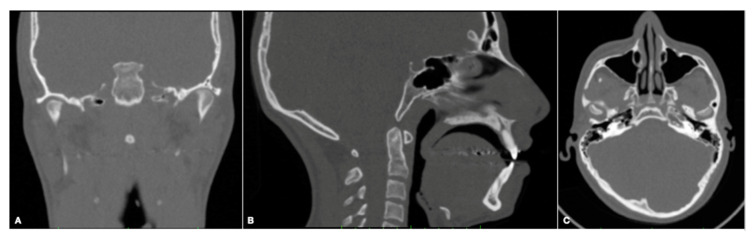
CT of a patient with left side condylar hyperplasia. (**A**) Coronal view. (**B**) Sagittal view. (**C**) Axial view.

**Figure 2 diagnostics-13-01034-f002:**
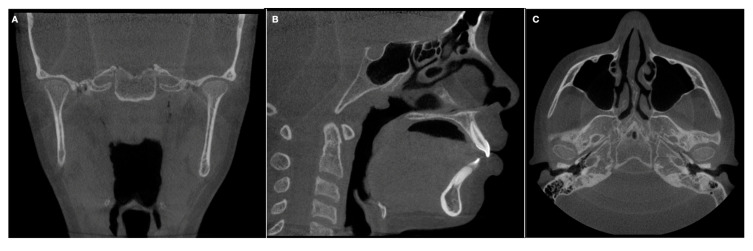
CBCT from a patient with no asymmetry (non-asymmetric group). (**A**) Coronal view. (**B**) Sagittal view. (**C**) Axial view.

**Figure 3 diagnostics-13-01034-f003:**
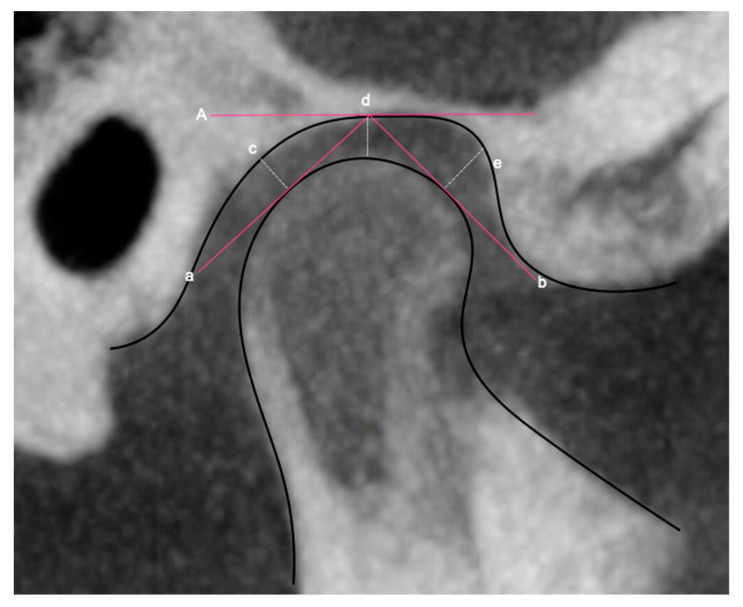
References for linear measurement of articular spaces (modified from Ikeda and Kawamura, 2009) [24]. **A.** True horizontal crossing the deepest point of glenoid fossa. **a.** Tangent line crossing the most posterior point of the mandibular condyle and the deepest point of the glenoid fossa. **b.** Tangent line crossing the most anterior point of the mandibular condyle and the deepest point of the glenoid fossa. **c.** Perpendicular from the most postero-superior point of the condyle on the tangent to the posterior wall of the glenoid fossa. **d.** Line from the most superior point of the mandibular condyle to the deepest point of the glenoid fossa. **e.** Perpendicular from the most antero-superior point of the condyle on the tangent to the anterior wall of the glenoid fossa. The distance in mm in these lines measures the anterior, middle and posterior articular space.

**Figure 4 diagnostics-13-01034-f004:**
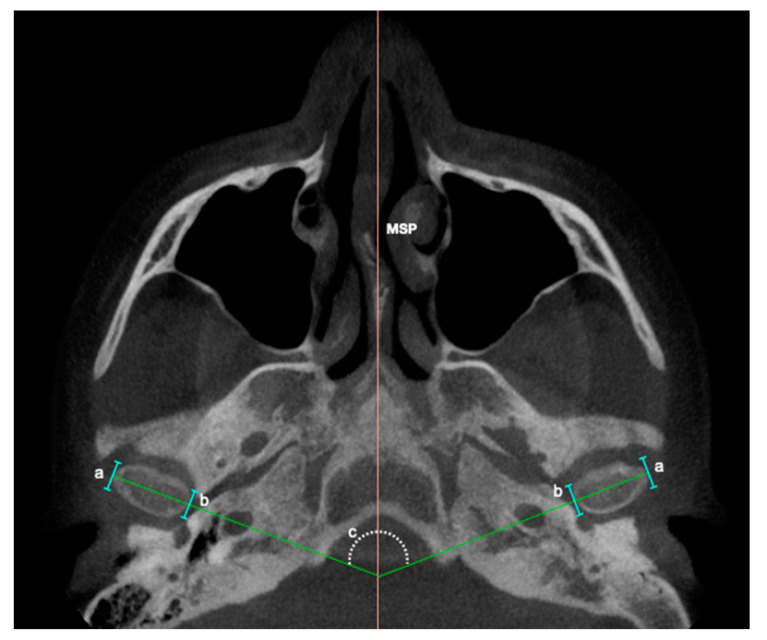
References to obtain the angular measurement of condylar axis with respect to the mid-sagittal plane (MSP). **a.** Lateral extreme of the mandibular condyle. **b.** Medial extreme of the mandibular condyle. **c.** Inner angle between MSP and the line drawn from the projection of higher mid-lateral length of each condyle.

**Table 1 diagnostics-13-01034-t001:** Description of the variables measured in CT and CBCT.

Variable		Description
**Articular space**	Posterior	Draw a tangent line to the posterior wall of the mandibular condyle. The most posterior-superior point of the condyle on the tangent line is located and from that point a perpendicular is traced to the posterior wall of the articular cavity. Data in mm. Figure 3c.
	Middle	Draw a line from the uppermost point of the mandibular condyle to the deepest point of the glenoid fossa. Data in mm. Figure 3d.
	Anterior	Draw a tangent line from the most anterior part of the mandibular condyle. The most antero-superior point of the condyle on the tangent line is located and from that point a perpendicular is traced to the anterior wall of the articular cavity. Data in mm. Figure 3e.
**Condylar Position**		Apply the equation: DC = (P − A/P + A) × 100%. DC (condylar displacement). P (posterior articular space). A (Anterior articular space). Method described by Pullinger and Hollender, modified by Pereira et al., 2007 [21,22].
**Condylar axis**		Inner angle between sagittal middle plane (SMP) and the line drawn from a projection of the highest middle-lateral length of each mandibular condyle. Figure 4.

**Table 2 diagnostics-13-01034-t002:** Intraobserver agreement for patients treated during the period 2015–2020 with CT indicated due to FA.

Variable	Measurement 1 (*n* = 20) *	Measurement 2 (*n* = 20) *	CCA **
Right joint space			
Anterior	1 (0.7; 1.2)	1 (0.7; 1.3)	0.89
Middle	1.2 (0.9; 1.9)	1.3 (0.7; 1.7)	0.93
Posterior	1.4 (1.3; 1.6)	1.5 (1.2; 1.7)	0.96
Left joint space			
Anterior	1.1 (0.9; 1.5)	1.3 (0.9; 1.6)	0.92
Middle	1.3 (1; 1.8)	1.2 (0.9; 1.7)	0.93
Posterior	1.4 (1.1; 1.8)	1.6 (1.2; 2)	0.92
Condylar axis			
Right	66.4 (60.5; 72.2)	66.7 (60.5; 72.4)	1
Left	65.4 (63.1; 74.1)	65.7 (63.8; 73.3)	1

* Median (p25; p75), ** Correlation coefficient of agreement.

**Table 3 diagnostics-13-01034-t003:** Demographic and clinical description of the patients with FA and subjects without FA or signs of TMD.

Variable (*n* = 153)	NA (*n* = 20)	MAP (*n* = 25)	GFA (*n* = 9)	FL (*n* = 8)	HE (*n* = 61)	HH (*n* = 11)	HF (*n* = 19)
Age *	22 (16; 29)	17 (13; 19)	14 (13; 16)	16 (14; 20)	17 (15; 24)	26 (17; 30)	23 (16; 30)
Gender **							
Male	8 (40)	12 (48)	2 (22.2)	3 (37.5)	24 (39.3)	2 (18.2)	8 (42.1)
Female	12 (60)	13 (52)	7 (77.8)	5 (62.5)	37 (60.7)	9 (81.8)	11 (57.9)
Affected side **							
Right	0 (0)	15 (60)	4 (44.4)	3 (37.5)	32 (52.5)	7 (63.6)	11 (57.9)
Left	0 (0)	10 (40)	5 (55.6)	5 (62.5)	29 (47.5)	4 (36.4)	8 (42.1)

NA: non-asymmetric, MAP: mandibular asymmetric prognathism, GFA: glenoid fossa asymmetry, FL: functional laterognathism, HE: hemimandibular elongation, HH: hemimandibular hyperplasia, HF: hybrid form. * Median (p25; p75), ** *n* (%).

**Table 4 diagnostics-13-01034-t004:** Condylar position in FA patients and non-asymmetric subjects without FA.

Condylar Position (%) (*n* = 153)	NA (*n* = 20)	MAP (*n* = 25)	GFA (*n* = 9)	FL (*n* = 8)	HE (*n* = 61)	HH (*n* = 11)	HF (*n* = 19)
Right side							
Anterior	7 (35)	16 (64)	7 (77.8)	6 (75)	27 (44.3)	10 (90.9)	12 (63.2)
Middle	11 (55)	8 (32)	2 (22.2)	2 (25)	19 (31.1)	0 (0)	5 (26.3)
Posterior	2 (10)	1 (4)	0 (0)	0 (0)	15 (24.6)	1 (9.1)	2 (10.5)
Left side							
Anterior	10 (50)	13 (52)	6 (66.7)	5 (62.5)	33 (54.1)	4 (36.4)	7 (36.8)
Middle	8 (40)	6 (24)	3 (33.3)	1 (12.5)	18 (29.5)	5 (45.5)	7 (36.8)
Posterior	2 (10)	6 (24)	0 (0)	2 (25)	10 (16.4)	2 (18.2)	5 (26.3)

NA: non-asymmetric, MAP: mandibular asymmetric prognathism, GFA: glenoid fossa asymmetry, FL: functional laterognathism, HE: hemimandibular elongation, HH: hemimandibular hyperplasia, HF: hybrid form.

**Table 5 diagnostics-13-01034-t005:** Comparison of condylar position between affected sides and kind of FA vs. without FA.

Diagnosis and Affected Side (*n* = 153)	Right Condylar Position (%)	*p* Value	Left Condylar Position (%)	*p* Value
NA	7.4 (−3; 17) *	Ref	11.7 (0.3; 24.1) *	Ref
MAP	15.7 (7; 33.3) *	0.04	12.5 (−4.3; 36.3) *	0.85
Right	29.5 (0.6; 35.8) *	0.16	-	-
Left	-	-	15 (−13.1; 33.2) *	1
GFA	25.5 (22.6; 28.4) *	0.06	24.3 (9.8; 50.4) *	0.13
Right	20.1 (-) **	0.24	-	-
Left	-	-	30.8 (24.3; 54.5) *	0.03
FL	19.9 (13.2; 29.8) *	0.08	24.3 (−2.6; 33.6) *	0.57
Right	22.8 (-) **	0.12	-	-
Left	-	-	27.5 (-) **	0.15
HE	8.7 (−9.9; 23) *	0.77	15.3 (−5.5; 30.5) *	0.93
Right	13.7 (0.6; 20.4) *	0.46	-	-
Left	-	-	15.3 (1.5; 30.6) *	0.64
HH	33.3 (25.3; 42.3) *	<0.01	7.1 (−6.8; 30.4) *	0.64
Right	33.3 (22.3; 40.4) *	0.03	-	-
Left	-	-	27.2 (-) **	0.48
HF	23.9 (2.3; 38.2) *	0.08	−1.7 (−12.4; 25.4) *	0.29
Right	18.2 (2.3; 36.4) *	0.23	-	-
Left	-	-	3.5 (−14.2; 15.6) *	0.3

* Median (p25; p75), ** Average. (For sample sizes < 5 Q range was not calculated.) NA: non-asymmetric, MAP: mandibular asymmetric prognathism, GFA: glenoid fossa asymmetry, FL: functional laterognathism, HE: hemimandibular elongation, HH: hemimandibular hyperplasia, HF: hybrid form. *n* (%).

**Table 6 diagnostics-13-01034-t006:** Comparison of condylar axis angle data by sides in without FA group vs. FA groups.

Diagnosis and Affected Side (*n* = 153)	Right Condylar Axis (°)	*p* Value	Left Condylar Axis (°)	*p* Value
NA	68.9 (61.3; 73.3) *	Ref	70.4 (64.7; 75) *	Ref
MAP	69.5 (63.8; 74.5) *	0.5	69.4 (64.9; 72.2) *	0.78
Right	70.7 (65.2; 76.3) *	0.27	-	-
Left	-	-	69.1 (67.1; 71.3) *	0.91
GFA	66 (62.9; 74.7) *	0.94	66.4 (63.5; 73) *	0.44
Right	63.7 (-) **	0.31	-	-
Left	-	-	67.7 (-) **	0.62
FL	63.6 (59.2; 72) *	0.5	66.9 (61; 73.7) *	0.6
Right	66.4 (-) **	0.9	-	-
Left	-	-	63.1 (-) **	0.57
HE	65.2 (59.5; 72.9) *	0.41	65.4 (61.8; 70.3) *	0.03
Right	69.1 (61.4; 75) *	0.57	-	-
Left	-	-	65.7 (62.7; 71.7) *	0.14
HH	60.6 (51.7; 65.5) *	0.06	56.5 (51.4; 66.1) *	<0.01
Right	61.3 (60.2; 69.5) *	0.53	-	-
Left	-	-	63.6 (-) **	0.27
HF	62 (55.1; 69.9) *	0.21	62.2 (55.6; 74.2) *	0.11
Right	65.7 (62; 73.8) *	0.92	-	-
Left	-	-	64 (60.2; 77.6) *	0.64

NA: non-asymmetric * Median (p25; p75), ** Average. (For sample sizes < 5 Q range was not calculated). MAP: mandibular asymmetric prognathism, GFA: glenoid fossa asymmetry, FL: functional laterognathism, HE: hemimandibular elongation, HH: hemimandibular hyperplasia, HF: hybrid form. *n* (%).

**Table 7 diagnostics-13-01034-t007:** Comparison of joint space data of without FA group vs. FA groups and sides.

Diagnosis and Affected Side (*n* = 153)	Right						Left					
	Anterior (mm)	*p*	Middle (mm)	*p*	Posterior (mm)	*p*	Anterior (mm)	*p*	Middle (mm)	*p*	Posterior (mm)	*p*
NA	1.9 (1.6; 2.3) *	Ref	2 (1.7; 2.3) *	Ref	2.8 (2.2; 3.6) *	Ref	2.7 (2.3; 3) *	Ref	2.1 (1.8; 2.8) *	Ref	2.6 (2; 3) *	Ref
MAP	0.9 (0.6; 1.3) *	<0.01	1.1 (0.8; 1.9) *	<0.01	1.5 (1.1; 1.8) *	<0.01	1.2 (0.9; 1.4) *	<0.01	1.5 (0.9; 1.8) *	<0.01	1.5 (1.2; 1.8) *	<0.01
Right	0.9 (0.6; 1.1) *	<0.01	1.1 (0.8; 1.9) *	<0.01	1.2 (1; 1.8) *	<0.01	-	-	-	-	-	-
Left	-	-	-	-	-	-	1.2 (1.1; 1.4) *	0.01	1.7 (1.5; 2) *	<0.01	1.6 (1.4; 2) *	<0.01
GFA	0.9 (0.8; 0.9) *	<0.01	1.3 (0.9; 1.5) *	<0.01	1.4 (1.3; 1.6) *	<0.01	1 (0.8; 1)	<0.01	1.6 (1.2; 2.3)	0.02	1.5 (1.4; 1.9) *	0.03
Right	1.2 (-) *	0.06	2 (-) **	0.13	1.8 (-) **	0.1	-	-	-	-	-	-
Left	-	-	-	-	-	-	0.8 (-) **	<0.01	1.5 (-) **	0.01	2.1 (-) **	0.1
FL	1 (0.7; 1.3) *	<0.01	1.3 (1.2; 1.6) *	<0.01	1.6 (1.4; 1.7) *	<0.01	1.2 (0.9; 1.8)	0.04	1.5 (1.1; 1.8) *	<0.01	1.9 (1.2; 2.2) *	0.02
Right	0.9 (-) **	0.01	1.2 (-) **	0.01	1.5 (-) **	<0.01	-	-	-	-	-	-
Left	-	-	-	-	-	-	1 (-) **	0.02	1.6 (-) **	0.01	2.1 (-) **	0.3
HE	1.1 (0.9; 1.4) *	<0.01	1.4 (0.9; 1.8) *	<0.01	1.3 (1.1; 1.7) *	<0.01	1.1 (0.9; 1.5) *	<0.01	1.4 (0.9; 1.7) *	<0.01	1.4 (1.2; 1.8) *	<0.01
Right	1 (0.9; 1.3) *	<0.01	1.2 (0.9; 1.7) *	<0.01	1.3 (1.1; 1.5) *	<0.01	-	-	-	-	-	-
Left	-	-	-	-	-	-	1.1 (0.9; 1.2) *	<0.01	1.1 (0.8; 1.5) *	<0.01	1.6 (1.2; 1.7) *	<0.01
HH	0.9 (0.8; 1.0) *	<0.01	0.8 (0.7; 2.0) *	<0.01	1.9 (1.3; 2.2) *	0.16	1.5 (1.1; 1.8) *	<0.01	1.2 (0.8; 1.5) *	<0.01	1.7 (1.4; 1.9) *	<0.01
Right	1 (0.9; 1.1) *	<0.01	1.2 (0.7; 2.4) *	0.01	2 (1.6; 2.2) *	0.26	-	-	-	-	-	-
Left	-	-	-	-	-	-	1.1 (-) **	0.03	0.9 (-) **	<0.01	2 (-) **	0.08
HF	1 (0.7; 1.2) *	<0.01	1 (0.8; 1.6) *	<0.01	1.5 (1.3; 1.7) *	<0.01	1.2 (0.9; 1.7) *	<0.01	1.3 (0.9; 1.7) *	<0.01	1.4 (1.1; 1.6) *	<0.01
Right	1 (0.8; 1.2) *	<0.01	1 (0.9; 1.6) *	<0.01	1.4 (1.3; 1.5) *	<0.01	-	-	-	-	-	-
Left	-	-	-	-	-	-	1.1 (1; 1.5) *	<0.01	1.1 (0.7; 1.6) *	<0.01	1.2 (1; 1.3) *	<0.01

NA: non-asymmetric, MAP: mandibular asymmetric prognathism, GFA: glenoid fossa asymmetry, FL: functional laterognathism, HE: hemimandibular elongation, HH: hemimandibular hyperplasia, HF: hybrid form. *n* (%), * Median (p25; p75), ** Average. (For sample sizes < 5 Q range was not calculated).

## Data Availability

Data available on request due to restrictions, e.g., privacy or ethical.

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
