# Peer review of "Positional Features of the Mandibular Condyle in Patients with Facial Asymmetry"

_diagnostics, 2023, doi:10.3390/diagnostics13061034_

Round 1

Reviewer 1 Report

The manuscript (diagnostics-2257686 ) entitled "Positional features of the mandibular condyle in patients with facial asymmetry" submitted to the journal Diagnostics describes a solid piece of scientific work which will be of interest for the general readership of this journal. However, a similar paper entitled "Positional change in mandibular condyle in facial asymmetric patients after orthognathic surgery: cone-beam computed tomography study (Choi et al. Maxillofacial Plastic and Reconstructive Surgery (2018) 40:13, https://doi.org/10.1186/s40902-018-0152-6 c)" is neither discussed nor cited. The authors should do this in order to improve their publication.

Author Response

Thank you so much for the review’s constructive comments.

The suggested article contributes considerably to the study carried out, therefore a paragraph in relation to it was added and placed in the discussion section, lines 256-258. Reference 39. The change is highlighted in yellow.

Reviewer 2 Report

The following points to be addressed by the authors before accepting the article for the publication

·       List of abbreviations needs to added. Example TMJ , TMD

·       How you justify with CT scans you can precisely define the accuracy of the results.

·       The slice thickness considered is 1 mm. But we can more detailed image if the slice thickness is 0.6mm. So how the authors justify their work.

·       Is there any significant difference are obtained with a change in slice thickness? Kindly compare one case, so you can justify your slice thickness.

·       Recent year’s reference needs to be added. Also, the introduction can be increased with the scope of the current work.

·       Can the obtained results can be validated against the published literature? This helps readers to understand the significance of the study.

·       Conclusion has to be rewritten with major findings of the work and also limitations of the current study can be added at the end of the discussion

Author Response

Thank you very much for the constructive feedback on the review. These have contributed to the strengthening of the manuscript and its presentation.

  1. We understand that there are many abbreviations, which is why we describe them in detail within the manuscript. Regarding the suggestion, the diagnostics journal template does not have a section for abbreviations and the editor's recommendation is to follow the journal template when submitting the article.
  2. Prior studies published by our group (Reference 31) cone beam images and clinical follow-up, allow us validation of these data in order to evaluate a definite diagnosis.
  3. A slice thickness of 0.75 mm has been used in this article, which is less than that used in previous articles that also evaluated joint and creneomandibular structures (reference 5, 1mm slice thickness and reference 23, 1.5mm slice thickness). In addition, this thickness has exquisite TMJ spatial resolution without increasing the radiation dose to patients associated with a smaller thickness. Changes in thickness do not significantly compromise the resolution of the TMJ structure and we have a policy of low dose radiation in young patients. Lines 96-97 of materials and methods. Highlighted in yellow.
  4. Answered in prior question.
  5. References 35 and 39 are added, which we consider current and with similar objectives to our research. They are in the Discussion section lines 250 and 256-258.
  6. The findings in each of the variables evaluated in the study have been contrasted with those of similar investigations.
  7. The conclusions have been rewritten, giving a logical order and highlighting the most relevant of the research. The discussion has increased and some potential limitations of the research have been mentioned. Lines 286-288. The changes are highlighted.

Round 2

Reviewer 2 Report

Can be accepted